# Identifying Interpretable Features in Convolutional Neural Networks

## Abstract

Single neurons in neural networks are often "interpretable" in that they represent individual, intuitively meaningful features. However, many neurons exhibit *mixed selectivity*, i.e., they represent multiple unrelated features. A recent hypothesis proposes that features in deep networks may be represented on non-orthogonal axes by multiple neurons, since the number of possible interpretable features in natural data is generally larger than the number of neurons in a given network. Accordingly, we should be able to find meaningful directions in activation space that are not aligned with individual neurons. Here, we propose (1) an automated method for quantifying visual interpretability that is validated against a large database of human psychophysics judgments of neuron interpretability, and (2) an approach for finding meaningful directions in network activation space. We leverage these methods to discover directions in convolutional neural networks that are more intuitively meaningful than individual neurons. In a series of analyses to understand this phenomenon we find, for instance, examples of *feature synergy* where pairs of uninterpretable neurons work together to encode interpretable features. These results contribute to a larger effort to automate interpretability research, providing a foundation for scaling bespoke perceptual judgments to the analysis of complex neural network models.

## 1 Introduction

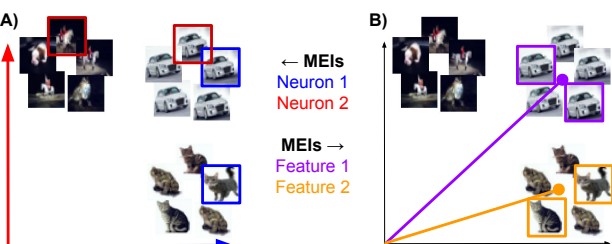

Figure 1: **Conceptual Overview. A**) A representation of two neuron's activations for different images. The highlights indicate maximally exciting images (MEIs) for each neuron. **B**) There exist directions in feature space that are more interpretable.

One of the oldest ideas in neuroscience is the *single neuron doctrine* (Barlow, 1972), i.e., the hypothesis that individual sensory neurons encode individually meaningful *features*.[1] The idea dates back to the early 1950s, when researchers began to find evidence of neurons that reliably and selectively fire in response to particular stimuli, such as dots on a contrasting background (Barlow, 1953) and lines of particular orientation and width (Hubel & Wiesel, 1959). These findings gave rise to the *standard model* of the ventral visual stream as a process of hierarchical feature extraction and pooling (Hubel & Wiesel, 1968; Gross et al., 1972; Riesenhuber & Poggio, 1999; Quiroga et al., 2005). Neurons in the early stages extract simple features, such as oriented lines, while neurons at later stages combine simple features to construct more complex composite features. In the highest stages, complex features are combined to yield representations of entire objects encoded by single neurons—the shape of a hand, or the face of a friend. This model has remained a dominant paradigm in sensory neuroscience for the last seven decades and ultimately inspired the development of convolutional neural networks (Fukushima, 1980).

---

[1]In this work, we adopt a pragmatic definition of *feature* based on human discernability, measured through psychophysics experiments (see below). For an attempt at a more formal definition see Elhage et al. (2022).

Mechanistic interpretability research, i.e., the single-unit electrophysiology of artificial neural networks, has yielded some remarkably coherent results, including many that align with classic empirical findings from neuroscience: the existence of neurons with center-surround receptive fields, color-contrast detectors, and oriented edge detectors that combine to form curve detectors in higher layers, for example (Olah et al., 2020). However, the study of individual neurons, both *in vitro* and *in silico*, faces two major problems. First is the inherent subjectivity of "interpretability," which generally necessitates the hand-inspection of neuron response properties. Second is the ubiquitous existence of hard-to-interpret units with mixed selectivity (Fusi et al., 2016; Olah et al., 2020).[2] We address both problems in this work by (1) defining a quantitative, automated measure of interpretability that does not rely on human inspection and (2) demonstrating a simple approach for finding meaningful directions in activity space that may not correspond to individual neurons.

Our first goal is to find a similarity metric that reflects human judgments of interpretability (Zimmermann et al., 2023). We then leverage this metric to test whether non-axis aligned directions in the neural activation space of CNNs trained on real data may be more interpretable than the neurons themselves—a test of the recently stated superposition hypothesis (Elhage et al., 2022). We compare simple low-level metrics (e.g., similarity in color) to mid-level metrics proposed to reflect human perceptual judgments (e.g., LPIPS (Zhang et al., 2018)) to high-level similarity metrics encoding object category (e.g., labels from a classification task) and assess our quantification of interpretability against a large-scale human psychophysics experiment conducted by Zimmermann et al. (2023). If a given metric provides a good match to human judgments of interpretability, it can serve as an automated interpretability tool: this allows for the quantification of interpretability in deep neural networks at scale (Leavitt & Morcos, 2020).

We perform a large number of analyses on a trained deep image model and find: directions that are more interpretable than individual neurons and exhibit sparse activations on the natural image manifold (Chen et al., 2018), and *feature synergy*—sparse combinations in activation space that yield more interpretable features than the constituent parts. Ultimately, our work pushes in the direction of automated interpretability research for CNNs, in line with recent efforts for language models (Bills et al., 2023; Cunningham et al., 2023; Gurnee et al., 2023; Sharkey et al., 2022).

## 2  METHODS

We propose an approach for quantifying the interpretability of neural network activations that is grounded in human judgement, yet is fully automated and scalable. In general, individual neurons — i.e., $N$ directions corresponding to basis vectors of an activation space $\mathbb{R}^N$ — might not be interpretable. Yet, other directions in $\mathbb{R}^N$ might be: we refer to them as *features*. For example, in Fig. 1 B) the human observer can define three directions that are interpretable and correspond approximately to horse-, car- and cat-like images. The superposition hypothesis stipulates that the activation space $\mathbb{R}^N$ of a neural network possesses several interpretable directions that are non-orthogonal (Elhage et al., 2022). Given a CNN, we aim to identify such directions and quantify their interpretability through the following three steps: **1. Collect neural network activations for a given dataset**. Images are passed through the network up to the layer under analysis, if this is a convolutional layer, we average across space as is common practice (Zimmermann et al., 2023). This generates a dataset in activation space $\mathbb{R}^N$; **2. Identify directions in activation space.** Directions may be provided by the neurons themselves (basis vectors) or by an algorithm (e.g., PCA, sparse coding, K-Means); **3. Quantify the interpretability of each direction**. We compute an interpretability index (II) as the average pairwise similarity of the top $M$ Maximally Exciting Images (MEIs, defined in the next subsection) for each direction. Through a suite of experiments, we argue that the II is a meaningful measure of the interpretability.

### 2.1  QUANTIFYING INTERPRETABILITY IN NEURAL NETWORKS

A neural network layer defines an activation space $\mathbb{R}^N$ with $N$ the number of neurons of that layer. We consider directions in this space, for example, individual neurons are represented as directions:

---

[2]One might wonder why evolution or gradient descent would be so kind as to make any neurons interpretable. Annecdotally, researchers have explained this as a result of the use of pointwise nonlinearities in deep networks. We provide a more formal argument for this explanation in Appendix B.

the basis vectors of $\mathbb{R}^N$. In activation space, some directions may be *interpretable*, in the sense that they detect a single feature or concept within the image data. For example, an interpretable direction may detect features such as edges, corners, textures in early layers, or more abstract patterns in later layers such as dogs, cats, trucks. By contrast, other directions respond to several unrelated features or concepts. For instance, Fig. 1 (A) shows the first neuron firing in response to unrelated car- or cat-like images.

Maximally Exciting Images (MEIs) are defined as synthetic images that maximally activate a given direction in activation space (Erhan et al., 2009). Given a direction $u$, we propose an **Interpretability Index (II)** computed as the average pairwise similarity of its top $M = 5$ MEIs:

$$\text{II}(u) = \frac{1}{M} \sum_{j=1}^{M} \sum_{k=1}^{M} \text{sim}\Big(x_j, x_k\Big). \tag{1}$$

In this work, we consider and compare several similarity metrics sim: color (low-level), LPIPS (mid-level) (Zhang et al., 2018), and category labels (high-level).

## 2.2 IMAGE SIMILARITY METRICS

We consider image similarity metrics that capture notions of similarity at different levels of abstraction: **1) Low-Level: Color** The color similarity between two images is defined as the difference between the average color in each image; **2) Mid-Level: LPIPS** Learned Perceptual Image Patch Similarity (LPIPS) (Zhang et al., 2018) is a perceptual metric used for assessing the perceptual differences between images. It relies on a CNN such as VGG or AlexNet that has been pre-trained on an image classification task. Given two images, LPIPS extracts their respective feature maps from multiple layers of the CNN. LPIPS then computes the distance between the corresponding feature maps. The distances are scaled by learned weights and then aggregated to yield a single scalar value representing the perceptual similarity between the two images; **3) High-Level: Labels** The label similarity between two images is a value equal to 0 if the two images have been assigned different labels during a reference classification task, and equal to 1 if the two images have been assigned the same label. In our experiments, we use the CIFAR dataset and associated classification task.

## 2.3 FROM HUMAN PSYCHOPHYSICS TO IN-SILICO PSYCHOPHYSICS

How can we validate whether the proposed interpretability index from Eq. 1 is indeed a sensible measure of interpretability? The concept of *interpretability* is intimately tied to human judgment. A long history of theoretical inquiry has demonstrated the impossibility of identifying necessary and sufficient conditions for many natural semantic categories (Stekeler-Weithofer, 2012). Due of this difficulty, we adopt a Wittgensteinian view, converting the question of whether a representation is interpretable into an empirical measure of the human interpretability judgment (Wittgenstein, 1953).

### HUMAN PSYCHOPHYSICS

Psychophysics is a field that quantitatively investigates the relationship between stimuli (e.g. images) and the perceptions they produce for human observers. Borowski et al. (2020) and Zimmermann et al. (2023) have demonstrated that large-scale psychophysics experiments can be leveraged for conducting quantitative interpretability research. In these works, researchers used the judgments of human participants to quantify the interpretability of neurons in trained artificial neural networks. In Zimmermann et al. (2023), participants are shown 9 minimally and 9 MEIs for a given neuron. The participant is then asked to select one of two query images $x_1, x_2$ that they believe also strongly activates that neuron (see App. D for an illustration). The *(human) psychophysics accuracy* obtained for that neuron is defined as the percentage of participants that are able to select the correct image.

### IN-SILICO PSYCHOPHYSICS

Psychophysics experiments provide a way of crowd-sourcing and quantifying human intuition of interpretability at scale. However, such experiments are time consuming, noisy and costly ($12,000 for (Zimmermann et al., 2023)). Here, we propose a method for automating psychophysics experiments, with a model that faithfully approximates human judgments while requiring no human input.

We replicate, *in-silico*, the experiments of Zimmermann et al. (2023), comparing different similarity metrics as proxies for human judgments. In our experiments, the model computes the maximum similarity, according to the image similarity metric, between each of the query images $x_1, x_2$ and the set of MEIs. The model then chooses as its response the image that is the closest to that set, i.e., that minimizes the cost:

$$C(x) = \text{sim}(x, \text{MEI}(u)) = \min_{k=1,\ldots,9} \text{sim}(x, x_k), \tag{2}$$

where $x_1, \ldots, x_9$ are the top 9 MEIs for a neuron or direction $u$, and sim the image similarity metric. The *psychophysics accuracy* for a given neuron or direction $u$ is defined as the percentage at which the model selects the correct query image for that neuron, i.e.:

$$\text{Acc}(u) = \frac{\text{\# of correct selections for direction } u}{\text{\# of queries with direction } u}. \tag{3}$$

We check in practice that directions $u$ with high interpretability index $\text{II}(u)$ are indeed more interpretable to a human observer. We note that we could have chosen the *in-silico* psychophysics accuracy $\text{Acc}(u)$ of direction $u$ to quantify its interpretability: the more interpretable $u$ is, the easier it is for participants to correctly select images associated with it (Zimmermann et al., 2023). However, we observe in practice that $\text{Acc}(u)$ is often at ceiling, and does not contain as much information as our proposed $\text{II}(u)$. Since it is also more expensive to compute, we use it only to validate the viability of the $\text{II}(u)$. Since the human psychophysics accuracy was computed only for directions corresponding to individual neurons, having in-silico psychophysics experiments is a key component of our approach. Note that we deliberately chose to work with MEIs rather than feature visualisations (Olah et al., 2017), because the latter showed consistently lower interpretability in different psychophysics studies (Borowski et al., 2020; Zimmermann et al., 2021; 2023).

## 3 RESULTS

### 3.1 COMPARISON: HUMAN VS. IN-SILICO PSYCHOPHYSICS

We first test the LPIPS similarity metric as a model of human perception of neuron interpretability. The experiments from Zimmermann et al. (2023) quantify the interpretability of neurons in models trained on ImageNet-1k through crowd-sourced human perceptual judgments. We reproduce this experiment *in-silico* by presenting the same image queries to a simple model based on the LPIPS metric. For each query, we evaluate whether the image selected by the model agrees with the image selected by human participants: it is a correct classification it they agree, incorrect otherwise. The results of this binary classification task are shown in Figure 2 for the LPIPS image similarity metric (Zhang et al., 2018), on five of its AlexNet layers.

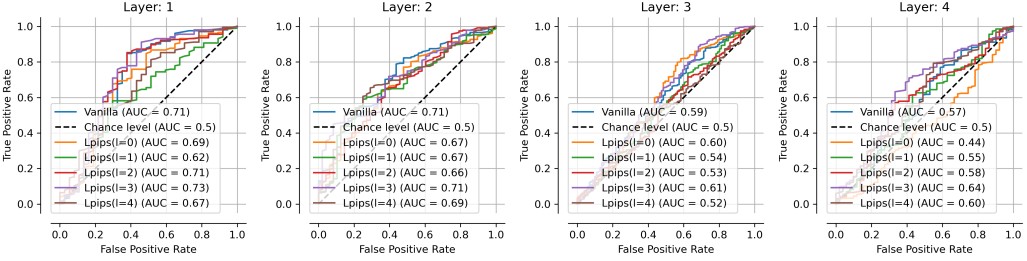

Figure 2: **Interpretability Metric vs. Human Behaviour.** Data from Zimmermann et al. (2023). Left to Right: Agreement between human and in-silico psychophysics on the predictability of the outputs of four layers within a ResNet50. Human and model agree on what makes a feature predictability for the ResNet50's early layers. For these layers, the proposed interpretability metric is a valid representation of the human's perception of interpretability. AUC: Area Under the Curve.

The predictions of the LPIPS model match human judgments well for earlier layers of the ResNet50—*layers* 1 and $2^3$—with an AUC up to 0.71 (Figure 2, left two panels). While there is certainly room for improvement, we conclude that this metric, based on LPIPS-based pairwise image comparison, serves as a good first proxy of human perception of interpretability. Crucially, our metric has the added benefit of not having to recruit a cohort of human participants. Thus, we will use this metric to evaluate the interpretability of features across neural network layers in the next subsections. Since the interpretability metric is more accurate for early layers, we focus the remainder of our analyses on layer 1 of the same ResNet50 architecture trained on CIFAR-10.

### 3.2 IDENTIFYING INTERPRETABLE DIRECTIONS IN FEATURE SPACE

We next apply the II to analyze the interpretability of features in a 50-layer ResNet50 pre-trained on the CIFAR-10 image dataset (Krizhevsky et al., 2009)[4]. We evaluate several methods for identifying interpretable directions in activation space: PCA, ICA, NMF, $K$-Means with cosine similarity, and the shallow sparse autoencoder used in Sharkey et al. (2022). We evaluate several similarity metrics and compute the II for each, comparing the interpretability of individual neurons in a layer with the interpretability of identified directions from that layer.

For a quantitative comparison, we present comparative box plots for the distributions of II indices for neurons and directions in Figure 3 while we vary: the LPIPS layer defining the similarity metric used in the II (left), the number of $K$ directions in activation space (middle), and distributions after splitting neurons into uninterpretable and interpretable groups (right). We observe that the $K$-means approach detects directions that are indeed more interpretable (higher II) than the individual neurons of the activation space — independently of the LPIPS layer considered for the II (Figure 3 left). Our method achieves higher II values (mean $= -0.0159$) than all baselines and the sparse autoencoder (best mean $= -0.0188$) (detailed comparison in App. A) [note that the II has arbitrary units]. Interestingly, the number of directions $K$ does not impact their IIs in the regime tested (Figure 3 left). Thus, we focus our analyses on the $K = N = 256$ setting for a fair comparison.

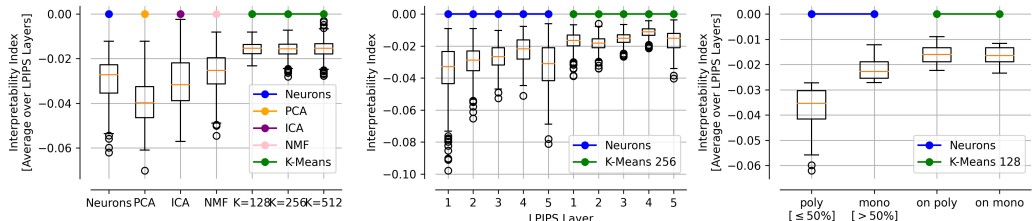

Figure 3: **Quantification of Interpretability.** Left: II score [a.u.] distribution for neurons ($N = 256$), PCA and ICA baselines, and K-Means as a function of $K \in \{128, 256, 512\}$. Middle: II score distribution for neurons ($N = 256$) and K-Means ($K = 256$) as a function of LPIPS layer. Right: II score distribution for uninterpretable neurons ($N = 128$), i.e. those with II score below the median, and interpretable neurons ($N = 128$), i.e., those with II score above the median; II score distribution for K-Means ($K = 128$) computed on each of these subsets separately.

The $K$-means approach can detect directions within subsets of uninterpretable neurons as well as within interpretable neurons, as we do not observe II differences in Figure 3 (right). Further, transforming interpretable neurons and uninterpretable neurons into their direction increases the II of both. We see a trend where the II increases with respect to the LPIPS layer used, which is a similar pattern as we saw in Figure 2. Like in Figure 2, using LPIPS Layer 3 yields the highest II.

For a qualitative comparison, Figure 4 shows the Maximally Exciting Images (MEIs) for 5 neurons (left) and 5 directions extracted from $K$-Means (right) selected in 5 different quantiles of II values (to avoid cherry picking in this qualitative comparison). The distributions of II indices is shifted towards the higher values for the directions detected by $K$-means, as shown by the II values associated with each quantile. This is confirmed by the visualization of MEIs which appear more visually coherent to the human observer for the directions (right) compared to the neurons (left).

---

[3]This refers to the PyTorch module names, corresponding to layers 10 and 23 in the network.
[4]Hosted at https://github.com/huyvnphan/PyTorch_CIFAR10

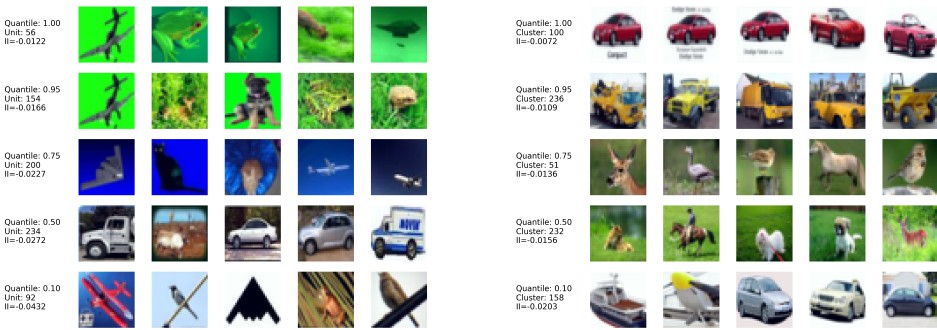

Figure 4: **MEIs of Neurons and Monosemantic Directions.** These are the Maximally Exciting Images (MEIs) for neurons (left) and directions (right) as retrieved with $K$-Means. To represent the interpretability index (II) distribution, we show neurons and directions at different II quantiles.

### 3.3 COMPARING SIMILARITY METRICS FOR IN-SILICO PSYCHOPHYSICS

We now compare the interpretability of the directions measured using the three image similarity metrics described in Section 2. Each metric defines similarity at a different level of abstraction, from low-level to high-level: same *color* (Figure 5 left), same *perceptual structure* as defined by LPIPS (Figure 5 middle) or same *category* (Figure 5 right). For each metric, we perform the *in-silico* psychophysics task from Section 3.1, varying the difficulty of the psychophysics experiment. The difficult of a task is by choosing query images that cause less extreme activations—i.e. are farther away from the set of MEIs (Borowski et al., 2020). This allows us to probe a more general understanding of the interpretability of a neuron or direction.

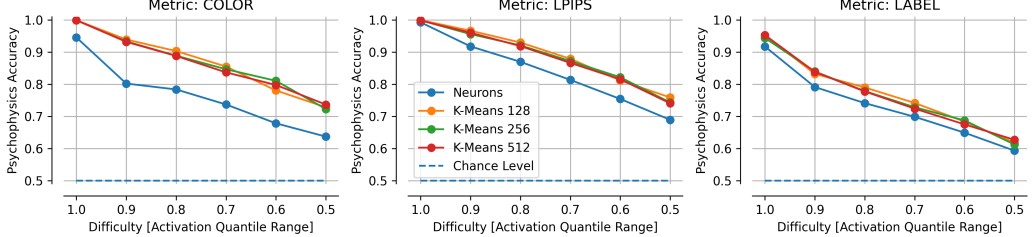

Figure 5: *In Silico* **Psychophysics Performance.** Accuracy across neurons and interpretable directions revealed by $K$-Means clusters ($K \in \{128, 256, 512\}$) for *in silico* psychophysics task for different levels of difficult, i.e., limiting query and reference image selection to the central range of activations (e.g., from the $0.45^{th}$ until the $0.55^{th}$ quantile, see (Zimmermann et al., 2023)). Predictions are made based on different metrics from low level semantics (colour match, left), over mid level semantics (LPIPS average over layers, center), to high level semantics (label match, right).

As expected, we see in Figure 5 that both the neurons and the directions have a decreased psychophysics accuracy as the task becomes more difficult. The directions detected by our approach are more predictable than the individual neurons across low, mid and high-level semantics and across task difficulties. The largest improvement over individual neurons is observed for the low-level semantics using colors, and the improvement decreases as we move towards higher level semantics. Additionally, as observed in Figure 3, the number of clusters $K$ does not impact the accuracy.

### 3.4 INTERPRETABLE NEURONS AS SPARSE ACTIVATIONS ON THE DATA MANIFOLD

One way of understanding interpretability is in terms of the distribution of a given neuron's activations over the image manifold. This concept and the following analysis are directly inspired by the *sparse manifold transform* (Chen et al., 2018). We take the top $M = 5$ MEIs for both neurons ($N = 256$) and K-Means ($K = 256$) features and compute all pairwise image similarities using LPIPS. We then embed this distance matrix into a 2D space for visualization purposes using t-SNE

(Van der Maaten & Hinton, 2008) (perplexity= 10) (Fig. 6). Each point in the visualization corresponds to a different image, and is colored according to a different scheme in each subplot. In Fig. 6:A, the color of each point indicates the average color of the image. In Fig. 6:B, color indicates the image label. In Figs. 6:E and 6:F, color indicates the activation of a single neuron (E) or $K$-Means feature (F) over the dataset. Activations for both neurons and $K$-Means features are computed as follows:

$$f_i(x) = e^{-\frac{d'}{\tau}}, \quad d' = \frac{d - \text{avg}(d)}{\text{s.d.}(d)} d = \|x - \mu_i\|_2^2 \tag{4}$$

Where $\mu_i$ is the location of the cluster centroids for $K$-Means features and a one hot vector for neurons. Taking the z-score before the exponential ensures a fair comparison. Finally, the temperature $\tau = 2$ is introduced for visualization purposes.[5]

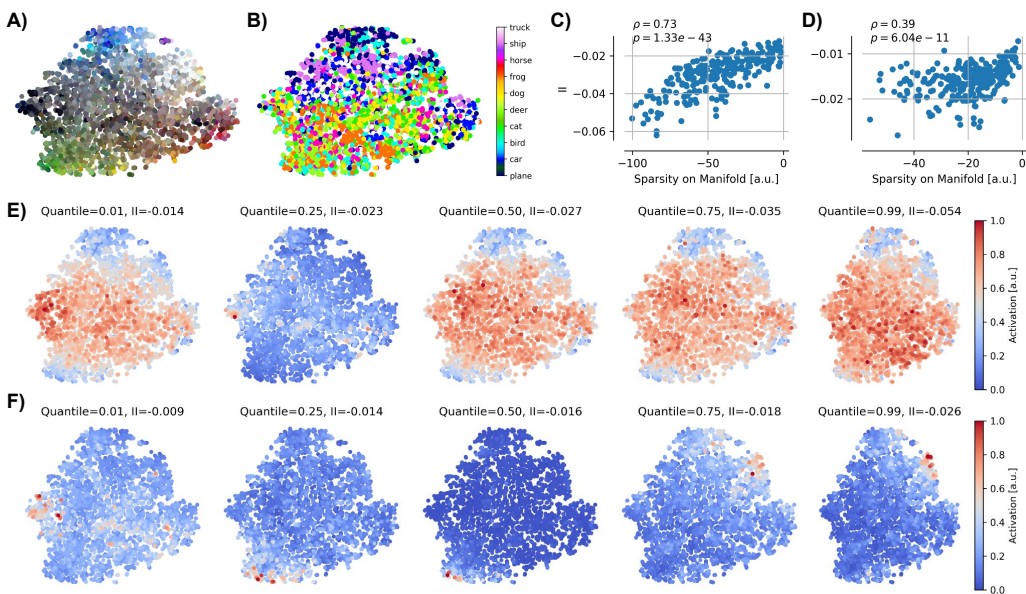

Figure 6: **Sparse Manifold Activations.** The natural image manifold (subset of MEIs) embedded in 2D with tSNE and coloured by average image colour **A)**, image label **B)**, activation of neurons **E)** and activation of features **F)**. There is a correlation between sparsity on the manifold (average distance of most activating points) and the II for both neurons **C)** and features **D)**.

We see that colour (Fig. 6:A) is a major factor in determining the layout of the manifold, and although labels tend to cluster locally, image category plays a lesser role (Fig. 6:B). Interestingly, we see that the features (Fig. 6:F) are much sparser on the manifold than the neural activations (Fig. 6:E). This suggests that more interpretable units are more sparsely active across the natural image manifold. Thus, we provide evidence (Fig. 6:C&D) for a long-standing hypothesis in the neural coding literature (Chen et al., 2018).

## 3.5 SENSITIVITY ANALYSIS

We investigate the sensitivity of the interpretable directions (directions), i.e., of the $K$-Means centroids. Specifically, we perturb each direction and quantify whether the perturbed direction is still interpretable. Our perturbation process is explained below. We interpolate from one $K$-Means centroid $\mu_a$ to another $\mu_b$ (and beyond to test extrapolation) and compute the II for these different directions in latent space. The intermediate directions are:

$$v(\alpha) = \alpha \mu_a + (1 - \alpha)\mu_b, \quad \text{for } \alpha \in \mathbb{R}. \tag{5}$$

We normalize each intermediate direction $v(\alpha)$ to maintain the same norm 1. For each direction $v$ along the interpolation path we compute the II index from the images that lie closest to that point in

---

[5]Lower $\tau$ would make Fig. 6:F look even sparser, and higher $\tau$ would make Fig. 6:E look even more uniform; using the same $\tau$ ensures a fair comparison.

latent space:

$$f_v(x) = -\|y(x) - v\|, \tag{6}$$

where $x$ is an image (an input), $y$ is the function representing the first layer of the CNN, $y(x)$ is the feature associated to image $x$ in latent space, and $\|\|$ is the Euclidean norm in latent space. The results are shown in Figure 7 B). We observe that $\mu_a$ and $\mu_b$, corresponding to interpolation factors $\alpha = 0$ and $\alpha = 1$ have higher II and that the II strongly drops for interpolating directions $v(\alpha)$. This supports the idea that the direction extracted via $K$-Means are interpretable. Moreover, based on signal detection theory (Dayan & Abbott, 2005), we hypothesize that more meaningful directions are so in virtue of being highly *selective* and less *sensitive* to input perturbations, i.e., to image perturbations Paiton et al. (2020). Thus, for each intermediate direction $v$, we also compute the norm of the input gradient:

$$\|\nabla_{f_v}|_x\| = \|\nabla[-\|y(x) - v\|]\| = \|\nabla[-\|y(x) - \alpha\mu_a + (1-\alpha)\mu_b\|]\|. \tag{7}$$

For a fixed intermediate direction $v$, this value quantifies how much the response of a feature $y(x)$ representing image $x$ changes given perturbations on image $x$.

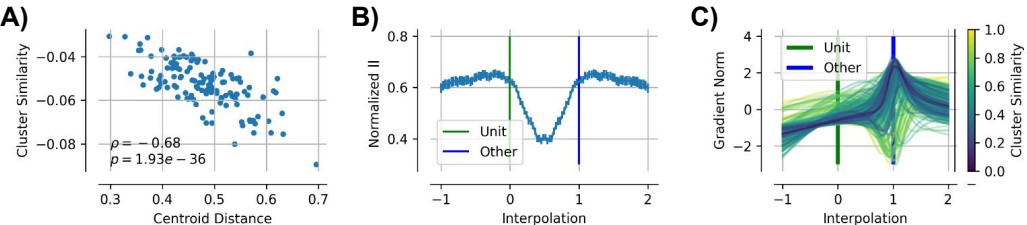

Figure 7: **Sensitivity Analysis. A)** Clusters that are further away from each other are have lower semantic similarity (measured as cross-II). **B)** Computing the II for interpolations (from one centroid to another) between and beyond all pairs of maximally separated clusters (determined with Hungarian algorithm). **C)** Sensitivity (i.e., average norm of the input gradient) for different interpolation points, coloured by the cluster similarity of start and end point in interpolation.

Figure 7 shows the average norm of the input gradient. We observe that the gradient's norm is generally much lower at a neuron's MEI ($\alpha = 0$) vs. the MEI of a different unit ($\alpha = 0$), unless they are very similar. We also find a weak but significant negative correlation (Spearman $\rho = -0.18$ $p < 3.110^{-5}$ between the interpretability and the minimal gradient norm, as shown in Figure 7 D). Together, these suggests that units which are more interpretable are also less sensitive to input perturbations at their preferred inputs. Consequently, a hypothesis derived from these analyses: neurons in CNNs that are more interpretable are also more robust to adversarial or noise perturbations.

### 3.6 PAIRWISE SYNERGIES BETWEEN NEURONS

Efficient coding principles such as minimal wiring length (Laughlin & Sejnowski, 2003), as well as the circuit analysis approach of mechanistic interpretability (Conmy et al., 2023; Nanda et al., 2023) inspire us to look for minimal subcircuits that increase interpretability. Specifically, we investigate the pairwise synergies between pairs of neurons. For all pairs of neurons $a, b$ in the same ResNet50 layer, we compute the II score for their added (z-scored) activity. The *synergy* is the difference between this II score and the maximum of their individual II scores to account for pairings with highly interpretable neurons:

$$\text{Synergy}(a,b) = \text{II}(a+b) - \max\left[\text{II}(a), \text{II}(b)\right]. \tag{8}$$

The synergy measures whether adding these neurons produces a direction in activation space that is more interpretable than taking each neuron individually. This is visualized in Figure 8 A) and B) which show two pairs of neurons $a, b$ with the highest synergy: the MEIs resulting from their addition are more interpretable that their individual MEIs.

The histogram of Figure 8 C) shows a large fraction of negative values of the synergy, i.e., most pairings are, as expected, detrimental for interpretability. However, a good fraction of the added neurons $a + b$ become more interpretable. Figure 8 D) shows that correlated neurons tend to have higher synergy but correlation alone does not explain everything: two neurons can be uncorrelated, yet their

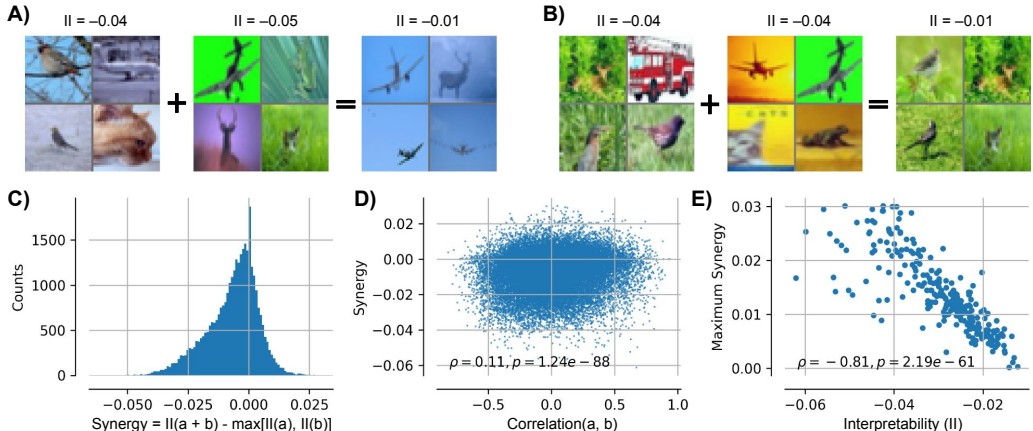

Figure 8: **Synergies. A), B)** Example pairs (two highest synergies) of neurons and the result when adding them (all visualized by their 4 MEIs). **C)** Histogram of synergies for every pair of neurons. **D)** A slight positive relationship between the correlation and synergy over all pairs of neurons (i.e., more correlated neuron pairs have higher synergies). **E)** A strong negative relationship between the II of a neuron and the maximum synergy it can achieve (i.e., pairings dilute interpretable neurons).

addition can produce a very interpretable feature. This shows that our notion of interpretability is distinct from the familiar notion of (un)correlation. Lastly, we find that more interpretable neurons (higher II) show lower maximal synergy (Figure 8 E)). This suggest that their representation is already interpretable and that any pairing would only dilute it.

## 4   DISCUSSION

In this work, we have proposed a quantitative metric of *interpretability* and a method for finding interpretable features in activation space. We hope that further research will find better metrics[6] and better feature identification methods. Nevertheless, we believe that our initial combination of metric and feature recovery method used here demonstrates the viability of our framework for automating interpretability research for vision models. In particular, we emphasize the value of validating quantitative metrics of interpretability against large-scale human psychophysics experiments of interpretability (Zimmermann et al., 2023). This allows us to scale human intuition to large-scale, complex neural network models—thus automating what is ordinarily done in mechanistic interpretability research by hand (Leavitt & Morcos, 2020). We hope that this approach will ultimately lead to a better understanding of neural coding principles and cast light into the black box of deep network representations.

*Mixed selectivity* is widely observed in neuroscience, (Rigotti et al., 2013) and there are coding advantages believed to be conferred by such a representation (Fusi et al., 2016)—for instance, in the case of *representational drift*, a phenomenon observed in cortex in which neurons change their tuning over time while maintaining a stable representation as a population (Rule et al., 2019; Driscoll et al., 2022; Masset et al., 2022). Moreover, such a code may be more robust to input perturbations, as suggested by our sensitivity analysis. In Appendix C, we show that network activations follow the same spectral power law as cortical representations (Stringer et al., 2019). That is, they are low-dimensional enough to maintain differentiability (i.e. they are robust to input perturbations), while being high-dimensional enough to capture the data structure. This suggests a *universal* coding strategy employed by biological and artifical neural networks alike. We believe that future analyses grounded in a quantified metric of interpretability may illuminate the computational function of these convergent neural coding strategies and more.

---

[6]Preliminary experiments with newer perceptual metrics (Fu et al., 2023) already show improved match to the human psychophysics data.

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

## 5    Appendix

## A    Comparison with a Sparse Autoencoder

We compare the K-Means approach for identifying interpretable directions in activation space to the shallow sparse autoencoder used in Sharkey et al. (2022). The model is a single-layer overcomplete (i.e. latent dimension is greater than input dimension) autoencoder trained with an L1 penalty on its hidden layer activation. We see in Fig. 9 that the directions identified by the sparse autoencoder are more interpretable according to our metric than the original neuron basis. However, we find that the K-Means approach performs better than the sparse autoencoder. In Sharkey et al. (2022), the authors assessed the relationship between source recovery and sparsity, using synthetic data containing known features. Here, we perform the same analysis, but with the II as a proxy for ground truth feature recovery. The functional relationship that we obtain between sparsity and II is remarkably similar Fig. 9 a&b, which suggests that the II may provide a good proxy for ground truth feature recovery.

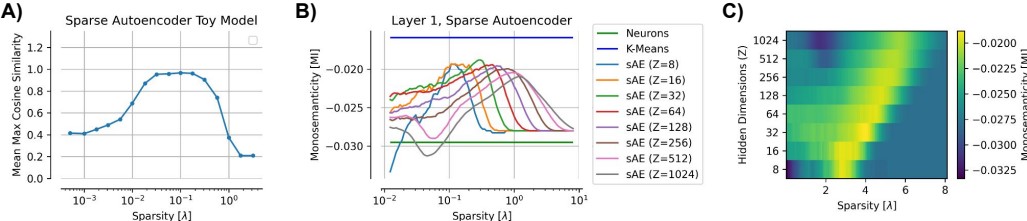

Figure 9:    **Model Comparison with Sparse Autoencoder. A)** Sparse autoencoder results reproduced from (Sharkey et al., 2022). **B), C)** We train the same sparse autoencoder model and measure the II index for different number of hidden dimensions ($Z$) and $L_1$ sparsity penalties ($\lambda$).

## B    Monosemanticity and the Privileged Basis Hypothesis

Why should we see individual neurons learning meaningful representations at all? Recent research in the mechanistic interpretability literature has suggested that there exists *privileged bases* in neural networks, corresponding to neurons, emerging from nonlinearities such as ReLU that operate per neuron. The intuition behind the privileged basis needs further explanation.

For $K$ neurons, there are $2^K$ quadrants as shown in Figure 10 A) for $K = 2$. We consider what happens for a feature vector in each of these quadrant after the application of ReLU. Of those, 1 (all positive) stays untouched, as shown in yellow in Figure 10 B. Another 1 quadrant (all negative) becomes zero: shown in purple. Next, $K$ quadrants get represented by 1 neuron (i.e. $K$ dimensions are collapsed to 1) shown in blue and green; $K - 1$ get represented by 2 neurons (i.e. $K$ dimensions are collapsed to 1), etc. In other words, each neuron participates in encoding points in $K!$ quadrants of compressed dimensionality. Now, we ask: which of these neurons should be more interpretable?

Consider data points unequally distributed into the different quadrants, with one quadrant (bottom right in Figure 10 A)) containing more points than another (top left in Figure 10 A)). Neurons in charge of representing the features from a quadrant with many point is more active. A resulting hypothesis is that neurons which are more active when others are inactive, i.e., which are active alone, are more interpretable — and form the priviledged basis.

Figure 10 C) uses our our interpretability index (II) to confirm this hypothesis by showing a negative correlation between a co-activation measure ("Active with Others") and II. The co-activation measure is defined as the correlation between each neuron's response and the average population response.

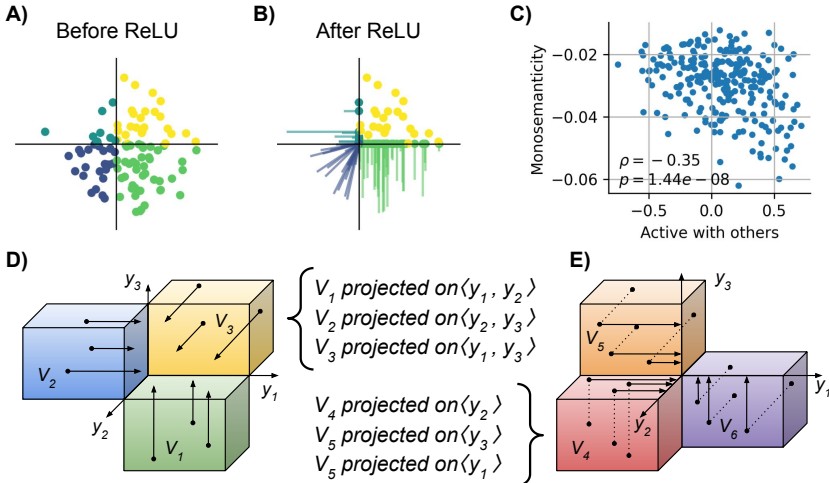

Figure 10: **Privileged Basis Hypothesis.** Activations before **A)** and **B)** after ReLU nonlinearity. In this scenario, the first neuron is more active as it has to represent the large number of (green) points in the bottom right quadrant, while the second neuron only needs to represent few (blue) points in the top left quadrant. **C)** Significant negative correlation between neurons that are more active together with others (measured as the correlation between the neuron's activity and the population, excluding the neuron, average) and the interpretability index (II). **D** In a three dimensional space, the positive only quadrant remains untouched by ReLUs, the negative only quadrant gets mapped to 0, three quadrants ($V_1, V_2, V_3$, **D**) are projected onto two dimensional subspaces, and three quadrants ($V_4, V_5, V_7$, **D**) are projected onto two one subspaces. Thus, e.g. neuron $y_1$ has to represent (together with $y_2$) all of $V_1$, and it also has to represent (completely on its own) all of $V_5$.

## C    DIMENSIONALITY OF THE NEURAL ACTIVATION MANIFOLD

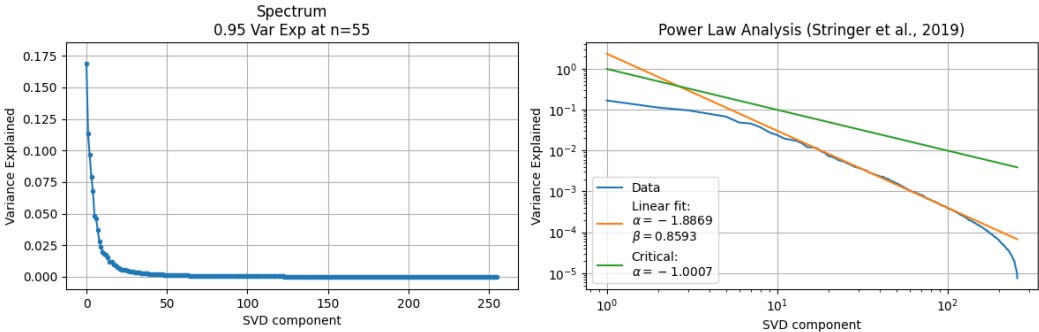

Figure 11: **High Dimensional Smooth Activation Manifold.** Same analysis as in Stringer et al. (2019), showing that activations in CNN feature space are high dimensional within the constraints of remaining differentiable. A spectrum that decays slower than the critical value (green line in right plot, Stringer et al. (2019)), would be non-differentiable and, therefore, highly non-robust. Remarkably, this spectral behaviour is the same as observed across many cortical areas.

## D    TASK EXPLANATION

Here is an illustration of the task used by Zimmermann et al. (2023).

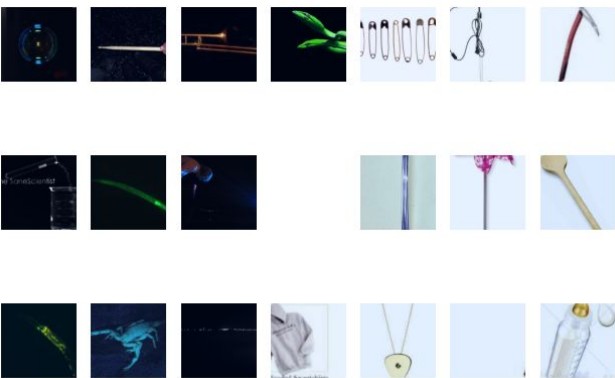

Figure 12: **Psychophysics Task.** Left images indicate positive reference ($MEI(y)$), right images indicate negative reference ($MEI(-y)$). The center images are the querries, the participant, here has to select the top image.

## E    ADDITIONAL COMPARISONS TO PSYCHOPHYSICS DATA

Here we look at some of the other exemplary models studied by Zimmermann et al. (2023).

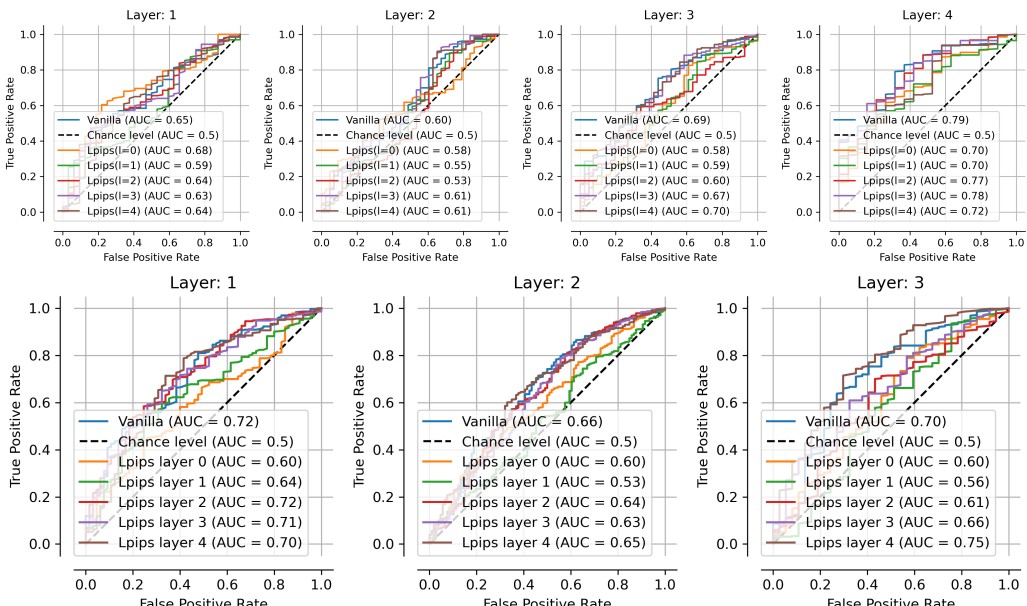

Figure 13: **Psychophysics Match for Other Models.** Data from Zimmermann et al. (2023). Top, Clip ResNet50; Bottom, GoogLeNet. Left to Right: Agreement between human and in-silico psychophysics on the predictability of the outputs of different layers in the network. Human and model agree on what makes a feature predictability for the early layers. For these layers, the proposed interpretability metric is a valid representation of the human's perception of interpretability. AUC: Area Under the Curve.

