# OpenReview forum: "Identifying Interpretable Features in Convolutional Neural Networks"
_ICLR.cc/2024/Conference — ICLR 2024 Conference Withdrawn Submission_

### Official Review · Reviewer_WTDd · 2023-10-27

**Soundness:** 3 good
**Presentation:** 3 good
**Contribution:** 3 good
**Rating:** 5
**Confidence:** 4

**Summary:**

This paper delves into focusing on developing an interpretable framework utilizing single neurons. The framework delves into utilizing interpretability but in an automated and scalable manner. They do this by collecting the neural network activations from the dataset, then identifying the directions in that space, and then quantifying the interpretability of the directions in that space. By doing this, they provide some metrics which are interpretability index (II) and maximally exciting images (MEI). They also use MEI and II to compare among other approaches. For instance, they apply II in a ResNet50 and compare against PCA, ICA, NMF, K-Means, and a sparse autoencoder to show why their approach is beneficial.

**Strengths:**

A good amount of experiments to compare human vs in-silico psychophysics. In addition, selectivity analysis is another good approach to test. From other papers that utilize interpretability, they sometimes do not show the sensitivity analysis, thus it is good to add.

The approach shows promise and a good amount of work has been put in.

**Weaknesses:**

Writing
In paragraph 3.1, in the text you write LPIPS on five of the AlexNet layers but in the caption of Figure 2, you mention four layers of ResNet50. Please fix this because it is a bit confusing.

You mention a single feature or concept, consider adding in citations that work on concepts since that can be more interpretable.

Please define your definition in the beginning. In this space, there are so many definitions and towards the end you mention the activation space. It will benefit to state that in the beginning.

For the figures please consider putting in a concluding statement, for example in Figure 7 with each subpart, you mention or describe the situation but it would benefit to draw your conclusion. For instance, in Figure 8 in E) you mention or conclude that there is a strong negative correlation but in c) you just mention a histogram but consider adding in a bit more saying a histogram but most of it lies in 0 and is not symmetric.

Experiments
Why was TCAV not compared because with your approach it could be another baseline to assess.

More experiments among different models and datasets would help reinforce the submission. With it being automated and scalable, having larger models such as ResNet152 or ResNet101 could help motivate your statement and make it stronger. Plus relating this framework to scenarios could help benefit it a lot. For instance when looking at the activation space, what happens if there is a certain bias, will the activation space be easy to identify. If there are classes that are similar like in the CUB-200 dataset, can you utilize this framework to identify it.

**Questions:**

Please refer to the weaknesses section.

---

### Official Review · Reviewer_KsZY · 2023-10-29

**Soundness:** 2 fair
**Presentation:** 1 poor
**Contribution:** 2 fair
**Rating:** 5
**Confidence:** 5

**Summary:**

The authors investigate whether the activation patterns of the activity of individual units in neural networks are "interpretable". Interpretability is itself an ambiguous concept, which moistly boils down to whether humans can use English words to more or less describe the activation patterns of unit activations. The authors propose a way of quantifying such interpretability based on human psychophyiscs experiments, and also a way of finding directions that show better interpretability indices in the network activation patterns beyond the activity of individual units.

**Strengths:**

The most exciting aspect of this paper is that it moves beyond the interpretability of the activations of individual units into the interpretability of dimensions that arise from population of units in the network. Arguably, this is a better way to think about interpretability. Individual units need not be directly interpretable, but ultimately, the network as a whole (or subsets of a network) need to produce emergent behaviors that are useful, and part of this usefulness means alignment with human behaviors, and human behaviors are often described with words.

**Weaknesses:**

The presentation is very poor.
Fig. 1. Unfortunately, the way the schematic is designed, the activity of individual units is highly interpretable, the dimensions add essentially nothing to the already existing clusters. What would be more useful is a schematic where the activity of individual units is not interpretable per se.

Fig. 2 is impossible to read. The legends cover the entire figure.

what makes a feature predictability --> what makes a feature predictable (perhaps)

From what little can be discerned from Fig. 2, it seems that layer 1 would be more "interpretable"? This seems highly counterintuitive.  It is very unclear what Fig. 2 intends to measure.

Where is the II score defined? For a paper focused on interpretability, it is intriguing that the basic metrics are not defined.

**Questions:**

Basics: define all terms, make sure that the figures are not covered by legends.

It would be especially useful to focus on other layers. The focus on layer 1 is strange and makes the whole work less relevant. In general, investigators tend to be particularly interested in the interpretation of activity across all layers or in higher layers that are more related to the decisions made by the network.

---

### Official Review · Reviewer_SvsS · 2023-10-30

**Soundness:** 1 poor
**Presentation:** 2 fair
**Contribution:** 1 poor
**Rating:** 1
**Confidence:** 3

**Summary:**

In this work, the human interpretability of a direction within a representational space is approximated by the average LPIPS similarity of its top 5 maximally exciting images (Eq. 1). This interpretability index ("II") is compared to existing psychophysical data from Zimmermann et al. (2023), finding an association (AUC ≈ 0.5-0.7) between the index's predictions and human judgments. Next, the study employs the new index to conduct in silico observations on the interpretability of directions obtained from different methods (e.g., PCA, ICA, k-means), finding a higher II for directions obtained by k-means. Additional analyses inspect the impact of task difficulty on simulated psychophysics and find a correlation between sparse activation and interpretability, reduced interpretability of interpolations between distinct k-means directions, and synergistic interpretability effects for a minority of the unit pairs.

**Strengths:**

* The goal of modeling human interpretability judgments is important. Specifically, establishing a good index of interpretability could facilitate the generation of representations engineered for interpretability.
* The paper examines a broad spectrum of relevant hypotheses regarding feature interpretability.

**Weaknesses:**

Unfortunately, this work does not achieve its stated goal. The study considers a single, heuristic index of interpretability. The index lacks theoretical motivation, and no empirical comparisons are made against alternative indices. While it makes sense that directions that produce perceptually similar MEIs are more interpretable, it is unclear whether this property captures human interpretability substantially. The more principled index of in silico accuracy is dismissed for being "often saturated" (a problem that can be solved by taking neural noise into account) and said to be expensive to compute, but the reasons for the high cost are not explained. Furthermore, the sole comparison to the human data (figure 2) provides no indication of the reliability bound of the human data (i.e., the "noise ceiling"). Therefore, we cannot estimate how good (or bad) the index is in capturing human judgments.

Subsequent sections of the paper employ the proposed, yet insufficiently validated, index as a dependent measure. Since no further human experiments are conducted to empirically test the reported findings (e.g., greater interpretability of K-means directions), it remains unclear whether these results apply to human interpretability or solely to the proposed index.

Regarding the presentation, the detail provided is insufficient for replicating the experiments. This issue is exacerbated by the absence of an associated code repository

**Questions:**

1. What is the motivation for using the particular proposed index?
2. Can the arbitrary units of II be mapped to meaningful human psychophysical quantities, such as d-prime?
3. How do natural and synthetic MEIs compare with respect to the resulting II? I think that this distinction is not discussed in the paper.

---

### Official Review · Reviewer_c2XN · 2023-11-01

**Soundness:** 3 good
**Presentation:** 4 excellent
**Contribution:** 2 fair
**Rating:** 5
**Confidence:** 5

**Summary:**

The paper focuses on identifying interpretable features encoded by individual neurons and features encoded by directions in the activation space (a vector of neurons). The main methodological contributions are:
- a method for automating the identification of meanings associated with individual neurons. In summary, the method identifies a group of maximally exciting images and computes their pairwise perceptual similarity (e.g., using LPIPS) to compute an interpretability index.
- methods for identifying meaningful directions within activation space using decomposition (PCA, ICA) and clustering (K means) methods and the proposed interpretability index.

The main findings point to the fact that directions are more interpretable than individual neurons.

**Strengths:**

- The paper is very well written and pedagogical in its approach. It also makes it interesting by making references to works in neuroscience and drawing parallels.
- Automating the psychophysics analysis of Zimmermann et al. using similarity metrics such as LPIPS is novel
- The sensitivity analysis is interesting, and is novel to my knowledge

**Weaknesses:**

1. The paper does not properly consider the literature on interpreting neurons in computer vision research.

Analyzing activations using highly activating images for interpreting CNNs has been around since the early work of Zeiler and Fergus [Zeiler, Matthew D., and Rob Fergus. "Visualizing and understanding convolutional networks." Computer Vision–ECCV 2014:]. More importantly, Network Dissection [Bau, D., Zhou, B., Khosla, A., Oliva, A., & Torralba, A. (2017). Network dissection: Quantifying interpretability of deep visual representations. In Proceedings of the IEEE conference on computer vision and pattern recognition] is ignored, where neuron activation alignment with various concepts is quantified. These works are among the cornerstone works that study the interpretability of neurons in cnns using highly activating images and are well established. Of course, I understand the minor differences in methodology and, in fact, appreciate them (also appreciate referring to work of Erhan and Yonsinski); however, failing to mention major relevant works in computer vision and only relying on recent mechanistic interpretability works in NLP would lead to community confusion and reinventing the wheel. We could list more papers trying to assign semantics to individual neurons following the many citations of these cornerstone works. This fact also reduces my trust in the authors, having done a comprehensive survey to identify if more similar work has already been done.

Moreover, there are also influential works in computer vision on identifying meaningful directions in the latent space. These works are also motivated based on the fact that semantics are not encoded in individual neurons, and Elhage et al. (2023) is not the first work on this topic. The most famous one is perhaps T-CAV [Kim, Been, et al., "Interpretability beyond feature attribution: Quantitative testing with concept activation vectors (tcav)." International conference on machine learning. PMLR, 2018.] and a recent one is [Crabbé, Jonathan, and Mihaela van der Schaar. "Concept activation regions: A generalized framework for concept-based explanations." Advances in Neural Information Processing Systems 35 (2022): 2590-2607.].

These are famous, relevant works, and I ask the authors not only to outline the differences in the rebuttal but to do a proper literature analysis of these works and their follow-ups to make the contributions clearer. I think the specific technical contributions can be useful to the community. However, they need to be clearly distinguished from existing established works. It also helps with the promotion and reception of the work.

2. Although I appreciate the analysis in various experiments, the main findings are not novel. It was already known that analyzing individual neurons is a crude approximation, and they need to be considered in tandem (TCAV, Concept Activation Regions, and the works of Olah). The authors mention that the experiments confirm these findings (from Elhage et al), which I appreciate. However, the novelty of amajor finding is missing. Moreover, the interpretability of neurons and latent directions have been studied before (e.g. in TCAV, NetworkDissection and many following works). The authors are encouraged to highlight their contributions with respect to the literature properly.

3. "we focus the remainder of our analyses on layer 1 of the same ResNet50 architecture trained on CIFAR-10." It is quite a well-known phenomenon in cans that early layers are basic feature detectors (edges, colors, Haar-like features). The analysis will be more interesting if done on deep layers. Specifically, it is interesting to know if superposition occurs less or more in deeper features. I encourage the authors to increase the scale of their experiments and do more analysis, to make the work stronger.

4. Although the metric shows whether the highly activating images are similar in terms of color or LPIPS, it does not reveal what the concept associated with the images is (except for the case of using the label). One can use existing literature to identify these concepts as well.

**Questions:**

I would appreciate it if the authors addressed the three first weaknesses. Specifically, if they can explain how to address the first one which is of paramount importance.